# Sodium Leak Channel in Glutamatergic Neurons of the Lateral Parabrachial Nucleus Modulates Inflammatory Pain in Mice

**DOI:** 10.3390/ijms241511907

**Published:** 2023-07-25

**Authors:** Lin Wu, Yujie Wu, Jin Liu, Jingyao Jiang, Cheng Zhou, Donghang Zhang

**Affiliations:** 1Department of Anesthesiology, West China Hospital, Sichuan University, Chengdu 610041, China; wulin_1998@163.com (L.W.); yujie-wu@outlook.com (Y.W.); scujinliu@gmail.com (J.L.); 18983476564@163.com (J.J.); 2Laboratory of Anaesthesia and Critical Care Medicine, National-Local Joint Engineering Research Centre of Translational Medicine of Anesthesiology, West China Hospital, Sichuan University, Chengdu 610041, China

**Keywords:** chemogenetics, glutaminergic neurons, lateral parabrachial nucleus, NALCN, inflammatory pain

## Abstract

Elevated excitability of glutamatergic neurons in the lateral parabrachial nucleus (PBL) is associated with the pathogenesis of inflammatory pain, but the underlying molecular mechanisms are not fully understood. Sodium leak channel (NALCN) is widely expressed in the central nervous system and regulates neuronal excitability. In this study, chemogenetic manipulation was used to explore the association between the activity of PBL glutamatergic neurons and pain thresholds. Complete Freund’s adjuvant (CFA) was used to construct an inflammatory pain model in mice. Pain behaviour was tested using von Frey filaments and Hargreaves tests. Local field potential (LFP) was used to record the activity of PBL glutamatergic neurons. Gene knockdown techniques were used to investigate the role of NALCN in inflammatory pain. We further explored the downstream projections of PBL using cis-trans-synaptic tracer virus. The results showed that chemogenetic inhibition of PBL glutamatergic neurons increased pain thresholds in mice, whereas chemogenetic activation produced the opposite results. CFA plantar modelling increased the number of C-Fos protein and NALCN expression in PBL glutamatergic neurons. Knockdown of NALCN in PBL glutamatergic neurons alleviated CFA-induced pain. CFA injection induced C-Fos protein expression in central nucleus amygdala (CeA) neurons, which was suppressed by NALCN knockdown in PBL glutamatergic neurons. Therefore, elevated expression of NALCN in PBL glutamatergic neurons contributes to the development of inflammatory pain via PBL-CeA projections.

## 1. Introduction

Inflammatory pain is a complex condition characterised by multiple mechanisms, such as hyperexcitability and/or sensitisation of primary nociceptive neurons or nociceptors and neural-immune-endocrine interactions [1,2,3,4]. Inflammatory pain is one of the major contributors to the health care burden and is associated with many chronic diseases [5,6]. Despite the high prevalence and severity of inflammatory pain, the medications currently available to treat inflammatory pain are unsatisfactory, such as nonsteroidal anti-inflammatory drugs (NSAIDs), opioids and some adjuvant medications [7]. The combination of different drugs with different mechanisms of action improves the therapeutic effect, but their use is limited due to their increased risk of side effects [8]. Clinically, the “pain ladder” therapy published by the WHO is suitable for the treatment of acute and chronic pain but has little relevance to inflammatory pain [9]. The underlying cause of the problem is that the exact molecular mechanisms associated with inflammatory pain are not fully understood and involve complex peripheral and central mechanisms [2]. Therefore, it is clinically important to investigate the molecular mechanisms mediating inflammatory pain and to develop novel therapeutic agents.

The parabrachial nucleus (PBN) is a heterogeneous nucleus located on the dorsal side of the pons around the superior cerebellar peduncle (scp) [10]. PBN is involved in regulating many physiological functions of the body, such as respiration, circulation and pain [11,12]. PBN consists of three distinct subnuclei, including PBL, PBM (parabrachial medial nucleus) and KFN (Kölliker-Fuse nucleus) [11]. PBL mainly receives nociceptive input from dorsal projection neurons in the spinal cord and is most relevant to pain regulation [13,14,15,16]. Yang et al. [17] reported that PBL is essential for the transmission of pain signals from the spinal cord to the substantia nigra of the reticular formation (SNR). SNR-projecting PBL neurons can be activated by noxious stimuli [17]. In fact, most neurons in the PBL are glutamatergic [11,18]. A previous study [15] found that spinal cord projections to PBL are strictly glutamatergic. Taken together, these studies suggest an important role for PBL glutamatergic neurons in pain regulation. However, the molecular mechanism by which PBL glutaminergic neurons are involved in inflammatory pain is not fully understood.

The sodium leak channel (NALCN) is an unselective cation background channel that regulates resting membrane potential and neuronal activities [4,19]. NALCN is associated with the regulation of many important biological functions, such as respiration, pain, and anaesthesia [4,19,20]. Ford et al. [21] demonstrated that NALCN in spino-PBN neurons is involved in the upstream transmission of nociception and therefore may affect pain perception. Moreover, Zhang and colleagues [19] found that elevated expression and function of NALCN in the peripheral dorsal root ganglion and spinal cord contributes to the development of both chronic constriction injury (CCI)-induced neuropathic pain and complete Freund’s adjuvant (CFA)-induced inflammatory pain in rodents [4], highlighting NALCN as a potential molecular target for the treatment of pain. However, whether NALCN in brain nuclei is implicated in the modulation of pain remains unclear.

Based on the above evidence, this study hypothesised that NALCN in PBL glutamatergic neurons is involved in the modulation of CFA-induced inflammatory pain.

## 2. Results

Chemogenetic inhibition of PBL glutamatergic neurons increases the pain threshold in mice.

Chemogenetic manipulations were used to clarify whether the activities of PBL glutamatergic neurons are involved in pain regulation (Figure 1A). First, we injected viruses encoding hM4D(Gi) receptors (AAV2/9-mCaMKIIa-hM4D(Gi)-mCherry-ER2-WPRE-pA) or its control (AAV2/9-mCaMKIIa- mCherry-WPRE-pA) into the bilateral PBL of mice (Figure 1B, left). Three weeks later, mCherry expression was observed in PBL (Figure 1B, right). Immunofluorescence staining of C-Fos was performed after 2.5 mg·kg^−1^ clozapine N-oxide (CNO) injection (Figure 1C,D). Compared with the control group, there was a significant decrease in the number of C-Fos-positive cells in the PBL of the hM4D(Gi) group (Figure 1E, 30.3 ± 6.1 vs. 15.4 ± 4.4; n = 10 in both groups, *p* < 0.001), suggesting that PBL glutamatergic neuronal activity was significantly inhibited after CNO injection.

Next, we measured the von Frey threshold and thermal withdrawal latency in mice under chemogenetic manipulations. We intraperitoneally injected 2.5 mg·kg^−1^ CNO or an equal volume of saline into the mice 1 h before behavioural testing, and then the mice were placed on a metal elevated net for acclimatisation. In the control group, there was no significant difference in the von Frey threshold (Figure 1F, 0.4 ± 0.2 vs. 0.4 ± 0.2 g; n = 10, *p* > 0.05) or thermal withdrawal latency (Figure 1G, 10.2 ± 1.2 vs. 11.2 ± 1.5 s; n = 10, *p* > 0.05) after intraperitoneal injection of saline or CNO into mice. However, after intraperitoneal injection of CNO, the von Frey threshold (Figure 1H, 0.3 ± 0.2 vs. 0.6 ± 0.2 g; n = 11, *p* < 0.001) and thermal withdrawal latency (Figure 1I, 10.1 ± 1.4 vs. 12.8 ± 1.3 s; n = 11, *p* < 0.001) were significantly higher in the hM4D(Gi) group compared with intraperitoneal saline injection. The von Frey threshold and thermal withdrawal latency in control and hM4D(Gi) mice did not change significantly after saline injection compared with the baseline (Appendix A).

Chemogenetic activation of PBL glutamatergic neurons decreases the pain threshold in mice.

Next, we chemogenetically activated PBL glutamatergic neurons (Figure 2A) by injecting AAV2/9-mCaMKIIa-hM3D(Gq)-mCherry-ER2-WPRE-pA or AAV2/9-mCaMKIIa-mCherry-WPRE-pA into the bilateral PBL of C57BL/6J mice (Figure 2B, left). The expression of mCherry was observed in PBL after 3 weeks (Figure 2B, right). Immunofluorescence staining of C-Fos (Figure 2C,D) showed a significant increase in the hM3D(Gq) group compared with the control group after intraperitoneal injection of CNO into mice (Figure 2E, 29.5 ± 8.2 vs. 61.4 ± 12.2; n = 10 in both groups, *p* < 0.001), suggesting that PBL glutamatergic neuronal activity was significantly activated by chemogenetic activation.

Then, we measured the von Frey threshold and thermal withdrawal latency of the mice. In the control group, the von Frey threshold (Figure 2F, 0.4 ± 0.2 vs. 0.4 ± 0.2 g; n = 10, *p* > 0.05) and thermal withdrawal latency (Figure 2G, 11.2 ± 1.6 vs. 11.5 ± 1.6 g; n = 10, *p* > 0.05) of mice did not change significantly after intraperitoneal injection of CNO compared with saline. In the hM3D(Gq) group, the von Frey threshold (Figure 2H, 0.4 ± 0.3 vs. 0.1 ± 0.1 g; n = 11, *p* < 0.001) and thermal withdrawal latency (Figure 2I, 10.3 ± 1.1 vs. 6.7 ± 1.5 s; n = 11, *p* < 0.001) were significantly decreased after intraperitoneal injection of CNO compared with saline. There was no significant difference in the von Frey threshold and thermal withdrawal latency in control and hM4D(Gi) mice after saline injection compared with the baseline (Appendix A).

Inflammatory pain excites PBL glutaminergic neurons.

To further explore whether inflammatory pain was associated with PBL glutamatergic neuronal activity, we used complete Freund’s adjuvant (CFA) to construct an inflammatory pain model in mice (Figure 3A). After CFA pain modelling on the left plantar of C57 BL/6J mice, we found that the von Frey threshold and thermal withdrawal latency were significantly lower compared with equivalent saline modelling (Figure 3B, n = 8–9, *p* < 0.001), indicating successful model establishment. Immunofluorescence staining of C-Fos in the PBL was then performed in saline- and CFA-modelled mice (Figure 3C,D). The results showed that C-Fos significantly increased in PBL after CFA modelling compared with saline (Figure 3E, 9.4 ± 2.2 vs. 19.8 ± 5.2; n = 9–10, *p* < 0.001).

Next, we explored the changes in neuronal activity in response to pain stimulation by measuring local field potentials (LFPs) in PBL (Figure 3F). The results showed that slow-delta oscillations did not change significantly before and after saline modelling (Figure 3G,I–K, n = 8, *p* > 0.05) but were significantly enhanced after CFA modelling (Figure 3H,L–N, n = 8, *p* < 0.05). However, theta, alpha and beta oscillations did not change notably in the saline and CFA groups (Figure 3G–N, n = 8, *p* > 0.05).

Knockdown of NALCN in PBL glutamatergic neurons alleviates CFA-induced pain.

To identify whether NALCN on PBL glutamatergic neurons is involved in pain regulation in mice, we specifically knocked down the expression of NALCN on PBL glutamatergic neurons (Figure 4A). First, we performed immunofluorescence staining of NALCN in the PBL after saline and CFA plantar pain modelling in wild-type mice (Figure 4B,C). The results showed that NALCN was significantly increased in the PBL of mice after CFA modelling compared with saline (Figure 4D, 20.6 ± 3.3 vs. 39.6 ± 5.4; *p* < 0. 001; n = 9 in both groups).

Then, to knock down NALCN in PBL glutamatergic neurons, we microinjected NALCN gene silencing virus (AAV2/9-mCaMKIIa-mCherry-mIRNAI (NALCN)) and its control virus (AAV2/9-mCaMKIIa-mCherry-mIRNAI (NC)) (Figure 4E, left). We confirmed whether the virus was correctly expressed in PBL after 3 weeks (Figure 4E, right). We also performed fluorescence quantification of NALCN in the control and NALCN knockdown groups of PBL using immunofluorescence staining to ensure that NALCN had been successfully knocked down in PBL in the knockdown group (Figure 4F–H, 23.4 ± 4.5 vs. 13.4 ± 3.2; n = 10 in both groups, *p* < 0.001). Meanwhile, we found that CFA-induced C-Fos expression was significantly reduced after CFA modelling in the NALCN knockdown group (Figure 4I–K, 35.3 ± 6.7 vs. 17.8 ± 5.2; n = 10 in both groups, *p* < 0.001).

Next, we conducted behavioural tests, including the von Frey threshold and thermal withdrawal latency, in the control and NALCN knockdown groups. The von Frey threshold (Figure 4L, n = 9–10, *p* > 0.05) and thermal withdrawal latency (Figure 4M, n = 9–10, *p* > 0.05) were not significantly different between the two groups of mice after saline plantar modelling. There was no difference in the basal values of the von Frey threshold (Figure 4L, 0.4 ± 0.2 vs. 0.3 ± 0.2 g; n = 8–11, *p* > 0.05) and thermal withdrawal latency (Figure 4M, 12.4 ± 1.6 vs. 12.1 ± 1.4 s; n = 8–11, *p* > 0.05) between control and NALCN knockdown group mice. However, after CFA modelling, the von Frey threshold (Figure 4L, n = 8–11, *p* < 0.05) and thermal withdrawal latency (Figure 4M, n = 8–11, *p* < 0.001) were significantly increased in the NALCN knockdown group compared with the control group during recovery. It returned to baseline levels faster than the control group.

Similarly, to further understand whether there were differences in PBL glutamatergic neuronal activity in control and NALCN knockdown mice before and after CFA pain modelling, we conducted local field potential analysis of PBL glutamatergic neurons. The results showed that slow-delta oscillations were significantly increased in control mice after CFA modelling (Figure 5A,C–E, n = 10, *p* < 0.05), but the NALCN knockdown group did not show such an increase (Figure 5B,F–H, n = 9, *p* > 0.05). Theta, alpha and beta oscillations were not significantly different in control and NALCN knockdown mice before and after CFA modelling (Figure 5E,H, n = 8–11, *p* > 0.05).

NALCN modulates CFA-induced pain via PBL glutamatergic neuron-central nucleus amygdala (CeA) projections.

Next, we used anterograde viral tracers and immunofluorescence staining to explore the downstream projections of PBL neurons involved in CFA-induced inflammatory pain (Figure 6A). First, we injected self-EGFP-labelled scAAV2/1-hSyn-EGFP-WPRE-pA anterograde tracer virus into the bilateral PBL (Figure 6B) and then observed each brain region 2 weeks later, after confirming the correct expression of the tracer virus in the PBL region (Figure 6C). We detected the projection fluorescence of PBL neurons in several brain regions, including the bed nucleus of the stria terminalis (BNST) (Figure 6D), central nucleus amygdala (CeA) (Figure 6E), central nucleus of the amygdala (PAG) (Figure 6F), and central nucleus of the inferior colliculus (CNIC) (Figure 6G).

To further identify the PBL downstream nuclei linked to pain transmission, we performed CFA plantar inflammatory pain modelling in wild-type mice. Immunofluorescence staining of C-Fos protein in the BNST (Figure 6H), CeA (Figure 6J), PAG (Figure 6L) and CNIC (Figure 6N) was performed after saline and CFA modelling. The results showed that C-Fos protein in the CeA was significantly increased after CFA modelling compared with saline (Figure 6K, 3.3 ± 1.4 vs. 8.1 ± 2.2; n = 9–10, *p* < 0.001). However, no significant difference in C-Fos protein was observed in the BNST (Figure 6I, 2.3 ± 1.3 vs. 2.8 ± 1.3; n = 9–10, *p* > 0.05), PAG (Figure 6M, 2.7 ± 1.3 vs. 3.1 ± 1.7; n = 9–10, *p* > 0.05) and CNIC (Figure 6O, 1.9 ± 1.3 vs. 2.1 ± 0.7; n = 9–10, *p* > 0.05) in CFA modelling compared with saline.

Subsequently, we conducted saline and CFA plantar pain modelling in PBL NALCN knockdown and control mice, followed by immunofluorescence staining for C-Fos protein in the CeA region of both groups (Figure 6P,R). The results showed that C-Fos was significantly increased in the CeA of the control group after CFA modelling compared with saline (Figure 6Q, 3.7 ± 1.4 vs. 7.7 ± 2.1; n = 10 in both groups, *p* < 0.001), whereas it did not change significantly in the NALCN knockdown group (Figure 6S, 3.5 ± 1.4 vs. 4.3 ± 1.4; n = 10 in both groups, *p* > 0.05). This suggests that NALCN knockdown in the PBL decreased neuronal activity in the CeA, which might be a downstream nucleus for PBL pain transmission.

## 3. Discussion

In this study, we showed that upregulated expression of NALCN in PBL glutamatergic neurons is involved in the development of inflammatory pain in mice. First, using chemogenetic inhibition or activation of PBL glutamatergic neurons, we demonstrated that PBL glutamatergic neurons were associated with sensory transduction in mice under physiological conditions. Then, we showed that PBL glutamatergic neuronal activity was enhanced when mice experienced inflammatory pain. To investigate the underlying molecular mechanism, we specifically knocked down NALCN in PBL glutamatergic neurons and found that NALCN knockdown alleviated CFA-induced pain. We further explored the downstream pathways of the PBL using antegrade virus tracing, which included the BNST, CeA, CNIC, and PAG, and finally showed that NALCN may modulate inflammatory pain via PBL glutamatergic neuron-CeA projections.

Roeder et al. [22] reported that activities of the rostral ventromedial medulla (RVM) evoked by nociceptive inputs were significantly attenuated by inactivation of PBL neurons. One study [18] showed that PBL sends excitatory glutamate to norepinephrine neurons in the A7 region of rats to participate in central analgesia and mediate pain inhibition at the spinal dorsal horn level. In the present study, chemogenetic activation or inhibition of PBL glutamatergic neurons reduced or increased basal pain thresholds, respectively, in mice, suggesting a close link between PBL glutamatergic neurons and pain sensation. However, one study [16] showed that chemogenetic inhibition of PBL*^Oprm1^* neurons did not change the thermal sensitivity of mice. Two factors may contribute to the inconsistent findings between their study and ours. First, a hot plate was used in their study [16] to measure the thermal sensitivity of mice, whereas we used an infrared heat pain tester. The two types of thermal pain measurement are not identical in manner or specific nature. Second, PBL*^Oprm1^* neurons represent only a portion of PBL neurons that express mu-opioid receptors. However, we manipulated all glutamatergic neurons in the PBL, which may generate a much larger effect. Therefore, PBL glutamatergic neurons other than *Oprm1^+^* neurons are suggested to affect sensory sensation.

Zhang et al. [19] reported that elevated NALCN expression and function in the DRG and spinal cord were associated with chronic constriction injury (CCI)-induced neuropathic pain. Li et al. [4] showed that NALCN in the DRG and spinal cord was also implicated in CFA-induced inflammatory pain. This evidence highlights the role of NALCN in the peripheral DRG and spinal cord in the regulation of chronic pain conditions. However, it is currently not known whether NALCN in brain nuclei and/or circuits regulates pain. The present study provided evidence that NALCN in PBL glutamatergic neurons modulated CFA-induced inflammatory pain.

Previous studies have shown that mutations in the NALCN gene are associated with many neurological diseases, such as infantile hypotonia [23,24], severe mental retardation [23,24], psychomotor retardation [24,25,26], and epilepsy [27]. Recently, one study showed that NALCN modulates inflammation-induced depression by maintaining the activity of glutamatergic neurons in the ventral dentate gyrus (DG) [28]. Future studies will investigate whether NALCN is associated with other inflammation-related neurological diseases.

The mechanisms by which NALCN modulates inflammatory pain are not fully understood. Evidence has shown that pain induces the release of substance P from injurious afferent nerve endings [29,30], which can activate NALCN currents in neurons by binding to tachykinin receptor 1 (TACR1) in a G protein-independent manner that requires the Src family of kinases (SFKs) and UNC80 [31,32,33]. A previous study also revealed that the cAMP/PKA signalling pathway may participate in the regulation of NALCN in neuropathic pain [19]. It will be interesting to explore the underlying pathways by which NALCN modulates inflammatory pain.

Previous studies have shown that PBN is involved in the regulation of various functions via different projection circuits [11,34]. For example, in the regulation of respiratory activity, the PBN projects to the nucleus tractus solitarius (NST), preBöt complex, rostral ventral respiratory group (rVRG), etc. [7,35]. In pain regulation, the PBN mainly projects to the CeA, BNST, PAG, intralaminar thalamic nucleus (ILN), ventral tegmental area (VTA), etc. [14,17,34,36]. Deng et al. [14] showed that PBN directly transmits injurious signals from the spinal cord to the intralaminar thalamic nuclei. One study [34] reported that different efferent neurons involved in pain regulation in PBL are associated with different components of pain regulation. For example, activation of efferent signals from the ventral medial hypothalamus (VMH) or the lateral periaqueductal grey matter (lPAG) drives avoidance behaviour, whereas activation of efferent signals from the ventral medial hypothalamus (VMH) to the BNST or CeA produces aversive memory [34]. Another study [17] also showed that a subpopulation of PBL neurons is essential for transmitting nociceptive signals from the spinal cord to the substantia nigra pars reticulata (SNR). In this study, we used cis-trans-synaptic tracer virus to bilaterally label the mouse PBL, and green fluorescence of the tracer virus was found at the CeA, BNST, PAG, and CNIC after 2 weeks. CeA, BNST and PAG are well-known downstream nuclei of PBL that have been reported in previous studies [14,34,36]. We further confirmed that NALCN knockdown in PBL glutamatergic neurons suppressed CFA-induced C-Fos expression in the CeA, suggesting that NALCN may modulate CFA-induced inflammatory pain via PBL glutamatergic neuron-CeA projections. Interestingly, we also found a small amount of tracer fluorescence in the central nucleus of the inferior colliculus (CNIC), which was not previously reported. The CNIC is a critical component of the auditory pathway that receives input from all inferior auditory nuclei located in the brainstem and projects upwards to the medial geniculate nucleus (MGN) of the thalamus [37]. However, C-Fos protein in the CNIC did not change before and after CFA pain modelling, suggesting that PBL-CNIC projections might not be involved in the regulation of CFA-induced pain. It will be interesting to determine their function in future studies.

Although only male mice were used in this study, we speculate that there might be no sex difference in the mechanisms of inflammatory pain associated with NALCN. In our previous studies, we showed no sex differences in either neuropathic [19] or inflammatory pain [4] associated with NALCN in the dorsal root ganglia (DRG) and spinal cord. In these two studies, we also showed that neuropathic or inflammatory pain could increase NALCN currents and neuronal excitability in both neonatal and adult rats. However, it is not clear whether this mechanism regulates inflammatory pain in aging rodents. Nevertheless, it will be interesting to confirm whether NALCN modulates inflammatory pain in a sex- or age-dependent manner in future studies.

There are several limitations in this study. First, NALCN is related to the regulation of several physiological functions, and it is not clear whether NALCN knockdown in PBL glutamatergic neurons causes other phenotypic abnormalities. Second, the present study did not examine the differentially expressed genes in NALCN knockdown mice. Examining differentially expressed genes would help to elucidate downstream pathways implicated in the diseases associated with NALCN. Finally, we did not apply patch-clamp recordings or in vivo electrophysiological techniques to explore the NALCN currents and activities of PBL glutamatergic neurons under CFA pain modelling.

In summary, our study uncovered a novel ion channel, namely, NALCN, that modulates inflammatory pain via PBL glutamatergic neuron-CeA projections. NALCN in PBL glutamatergic neurons may be an underlying therapeutic target for the control of inflammatory pain.

## 4. Materials and Methods

### 4.1. Animals

C57BL/6J male mice (8 weeks, 20–22 g) were given free access to food and water and housed under standard conditions (22–24 °C, 45–55% humidity) with a 12:12 light cycle in this study. All procedures were approved by the Animal Ethics Committee of the West China Hospital of Sichuan University (Approval ID: 2021420A, Chengdu, Sichuan, China) and complied with the Animal Research guidelines: Reporting of In Vivo Experiments (ARRIVE) guidelines. All efforts were made to reduce the number of experimental animals and their suffering as much as possible.

### 4.2. Stereotaxic Injection

Wild-type mice were randomly grouped and labelled with ear tags. Mice were anaesthetised with 2% isoflurane and fixed in a stereotaxic device (RWD, Life Science, Shenzhen, China). Their body temperatures were maintained by a heating blanket. After cutting the scalp to expose the skull of the mice, two holes were drilled into the skull above the PBL to inject virus into the bilateral PBL (bregma: −5.4 mm, lateral: ±1.2 mm, depth: −3.4 mm). Viruses were injected at a rate of 100 nL/min, and the total volume was 500 nL except for tracer virus (300 nL). In chemogenetic manipulation, AAV2/9-mCaMKIIa-hM3D(Gq)-mCherry-ER2-WPRE-pA,_AAV2/9-mCaMKIIa-hM4D(Gi)-mCherry-ER2-WPRE-pA_and AAV2/9-mCaMKIIa-mCherry-WPRE-pA virus were bilaterally microinjected into the PBL of C57BL/6J mice. To manipulate NALCN, we injected AAV2/9-mCaMKIIa-mIRNAI(NALCN)-mCherry-WPRE-pA_and_AAV2/9-mCaMKIIa-mCherry mIRNAI(NC)-WPRE-pA into the bilateral PBL. To study PBL projections, scAAV2/1-hSyn-EGFP-WPRE-pA anterograde tracer virus was used. After viruses were injected, the glass electrode remained in place for five minutes and then was slowly withdrawn to prevent virus reflux. After this procedure, the scalps of the mice were sutured and the mice were placed on a heating blanket for resuscitation. When the mice awakened, they were put back into their cage.

### 4.3. Immunofluorescence Staining

The mice were anaesthetised with 2% isoflurane and transcardially perfused with phosphate-buffered saline (PBS, pH 7.4) and 4% paraformaldehyde until their limbs stiffened. The brains were fixed in 4% paraformaldehyde for at least 24 h at 4 °C and then immersed in 30% sucrose solution for another 24 h. A freezing microtome (CM1850, Leica, Hesse-Darmstadt, Germany) was used to prepare 20-µm-thick brain slices.

Then, brain slices were labelled with primary antibodies at 4 °C overnight, including C-Fos (1:500, mouse, GTX16902, Neobioscience, Shenzhen, China), NALCN (1:500, rabbit, ASC-022, Alomone, Jerusalem, Israel), and C-Fos (1:500, rabbit, ab222699, Abcam, Cambridge, UK). After overnight incubation, the primary antibody was washed with PBS and incubated with the secondary antibody, including Alexa Fluor 488 goat anti-mouse (1:500, ZF-0512, ZSGB-BIO, Beijing, China), Alexa Fluor 555 donkey anti-mouse (1:500, A-31570, Thermo Fisher Scientific, Waltham, Massachusetts, USA) and Alexa Fluor 550 goat anti-rabbit (1:500, 550045, ZEN-BioScience, Chengdu, China) for 2 h. All images were taken with a Zeiss Axio Imager Z.2. and analysed using ImageJ software V1.8.0 (National Institutes of Health, Bethesda, MA, USA).

### 4.4. In Vivo Chemogenetic Manipulation

For behavioural experiments, clozapine N-oxide powder (CNO, Sigma, St. Louis, MO, USA) was diluted to 0.5 mg·mL^−1^ with 0.9% saline and injected intraperitoneally at 2.5 mg·kg^−1^. In the control group, CNO was replaced with an equal volume of saline. Behavioural experiments began 50–60 min after CNO or saline injection.

### 4.5. Behavioral Tests for Pain

Pain tests included the von Frey filaments and Hargreaves tests to evaluate mechanical and thermal sensitivities separately. Before the formal test, the mice were acclimated on a wire mesh elevated frame for 2 h a day for a week. On the experimental day, mice were placed individually in a transparent cube Plexiglas chamber (length = 10 cm) and habituated for 1 h until they were immobile but awake. For the von Frey test, an up-down approach was used [38]. In the Hargreaves test, mice lifted their feet after a thermal radiation beam (35% intensity) was emitted by an infrared heat pain tester (37570-001, Ugo Basile, Italy). The latency time between the start of the radiation and the mice lifting their feet was defined as the thermal withdrawal latency. If the withdrawal latency exceeded 20 s or the mouse moved autonomously, the data were discarded.

### 4.6. Inflammatory Pain Mode

Under 2% isoflurane anaesthesia, 20 μL CFA (1 mg·mL^−1^, Sigma-Aldrich, St. Louis, MO, USA) was injected subcutaneously with an insulin needle in the left plantar of mice. Control mice were injected with an equal volume of saline.

### 4.7. Implantation of Local Field Potential (LFP) Electrodes

To record the PBL local field potential, three electrodes were implanted into each mouse, including one recording electrode, one common electrode, and one grounding electrode. The grounding electrode can prevent interference from surrounding information during recording. The common electrode and grounding electrode were anchoring screws (1 mm diameter, 2 mm long) fixed to the skull, and the recording electrode was an insulating silver wire inserted into the PBL. Under 2% isoflurane anaesthesia, the mouse skull was fully exposed after cutting the scalp. When recording the PBL local field potential, the recording electrode was implanted into the left PBL, the common electrode was implanted into the right PBL, and the grounding electrode was implanted next to the bregma point (bregma: −1 mm, lateral side: −1.2 mm). The three electrodes were all connected to a mini plug and fixed with glass ionomer cement. When the mice awakened, they were placed back into the cage.

### 4.8. LFP Recordings and Analyses

The LFP recordings were conducted one week after surgery, and signals were recorded at a sampling frequency of 500 Hz using a Pinnacle EEG Recording System (Part# 8200-SL; Pinnacle Technology, Salinas, CA, USA). After original signals were obtained using the Sirenia acquisition system, MATLAB (version 2006a; MathWorks, Santa Clara, CA, USA) and GraphPad Prism 9.0 software (GraphPad Software, San Diego, CA, USA) were used to amplify, digitise and further analyse the signals [39]. A single recording duration was 40 min. The data from the 30–35 min period were selected for analysis and sample presentation.

### 4.9. Study Design

Part 1: First, we randomly and equally divided 24 C57BL/6J mice into two groups and microinjected AAV2/9-mCaMKIIa-hM4D(Gi)-mCherry-ER2-WPRE-pA and AAV2/9-mCaMKIIa-mCherry-WPRE-pA viruses. After approximately 3 weeks, 10 in the control group and 11 in the hM4D group were used for pain behavioural tests. We first tested the basal values of the von Frey threshold and thermal withdrawal latency in both groups. After intraperitoneal injection of saline or CNO, the von Frey threshold and thermal withdrawal latency were tested again (Figure 1A).

Part 2: In this part, we randomly and equally divided 24 C57BL/6J mice into two groups, and then AAV2/9-mCaMKIIa-hM3D(Gq)-mCherry-ER2-WPRE-pA and AAV2/9-mCaMKIIa-mCherry-WPRE-pA viruses were injected into the bilateral PBL of the mice. After waiting for 3 weeks, the basal values of the von Frey threshold and thermal withdrawal latency were tested in 10 mice in the control group and 11 mice in the hM3D group before applying treatment measures. Then, after intraperitoneal injection of saline or CNO, the von Frey threshold and thermal withdrawal latency were tested again in both groups (Figure 2A).

Part 3: First, we randomly and equally divided 20 C57BL/6J mice into saline and CFA groups (10 mice per group). The von Frey threshold and thermal withdrawal latency were tested on days 0 (basal value), 1 (saline or CFA plantar modelling), 3, 5 and 7. On day 0, basal values of the local field potential of PBL were tested in both groups. On day 1, the local field potential of PBL was tested in saline- or CFA-modelled mice (Figure 3A).

Part 4: Twenty C57 BL/6J mice were randomly and equally divided into saline and CFA groups (10 mice per group). The brains of the two groups were extracted 30 min after saline or CFA plantar modelling. Immunofluorescence staining of NALCN was performed in PBL. In addition, 48 C57 BL/6J mice were randomly and equally divided into 4 groups (12 mice per group); 2 groups of mice were injected with AAV2/9-mCaMKIIa-mIRNAI(NALCN)-mCherry-WPRE-pA virus into PBL, and another 2 groups were injected with AAV2/9-mCaMKIIa-mCherry mIRNAI(NC)- WPRE-pA virus. After waiting for 3 weeks, 4 groups of mice were tested for von Frey threshold and thermal withdrawal latency on days 0 (basal values), 1 (saline or CFA plantar modelling), 3, 7, 10, and 14 (Figure 4A).

Part 5: First, scAAV2/1-hSyn-EGFP-WPRE-pA anterograde tracer virus was microinjected into PBL of 5 C57 BL/6J mice. After waiting for 2 weeks, the whole brains of mice were removed and observed for tracer virus projection fluorescence. After determining the projection nuclei, twenty C57 BL/6J mice were randomly and equally divided into saline and CFA groups (10 mice per group). Whole brains of mice were taken 30 min after saline and CFA plantar modelling in the two groups. Immunofluorescence staining for C-Fos was performed on the identified projection nuclei of each mouse. Next, 40 C57 BL/6J mice were randomly and equally divided into 4 groups: 2 groups were injected with AAV2/9-mCaMKIIa-mIRNAI(NALCN)-mCherry-WPRE-pA virus, and another 2 groups were injected with AAV2/9-mCaMKIIa-mCherry mIRNAI(NC)-WPRE-pA virus. After waiting for 3 weeks, the 4 groups underwent saline or CFA plantar modelling followed by C-Fos immunofluorescence staining (Figure 6A).

### 4.10. Statistical Analysis

Data are expressed as the mean ± standard deviation (SD) or mean ± standard error of the mean (SEM), and statistical analyses were conducted using SPSS version 25.0 (SPSS Inc., Chicago, IL, USA) and GraphPad Prism 9.0 software (GraphPad software, San Diego, CA, USA).

The normality of data distribution was assessed by the Shapiro–Wilk test. Data from two groups were compared using independent samples *t* tests. Repeated measurement data were analysed using repeated-measures multivariate ANOVA. After immunofluorescence staining, data from mice without corresponding viral expression in PBL were excluded. Detailed statistical analysis methods are described in the figure legends. *p* < 0.05 was considered statistically significant.

## 5. Conclusions

In conclusion, this study demonstrated that NALCN in PBL glutamatergic neurons is a key ion channel involved in the regulation of inflammatory pain. NALCN may serve as a critical molecular target for the control of inflammatory pain.

## Figures and Tables

**Figure 1 ijms-24-11907-f001:**
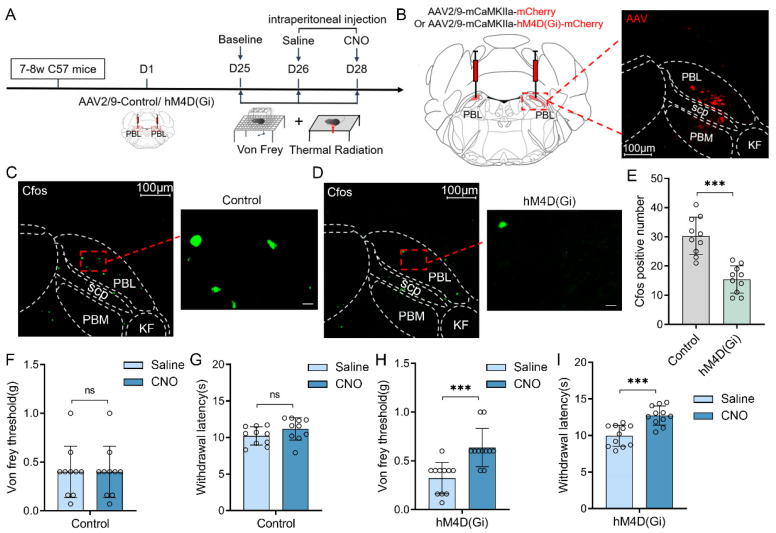
Chemogenetic inhibition of PBL glutamatergic neurons increases the pain threshold in mice. (**A**) Flow chart of the behavioural measurements in control and genetically manipulated mice. (**B**) Schematic of virus injection and hM4D(Gi) and mCherry expression in PBL glutamatergic neurons. Scale = 100 μm. (**C**,**D**) C-Fos immunofluorescence (green) in PBL after CNO injection in control (**C**) and genetically manipulated mice (**D**). Scale = 100 μm (**left**) or 10 μm (**right**). (**E**) C-Fos count comparison after CNO injection in control and hM4D(Gi) mice after immunofluorescence staining (n = 10 in both groups). (**F**,**G**) Von Frey threshold (**F**) and withdrawal latency (**G**) in control mice after saline or CNO injection (n = 10). (**H**,**I**) Von Frey threshold (**H**) and withdrawal latency (**I**) in hM4D(Gi) mice after saline or CNO injection (n = 11). Data are presented as the mean ± SD. *** *p* < 0.001, ns: no significance by a two-tailed independent samples *t* test (**E**–**I**).

**Figure 2 ijms-24-11907-f002:**
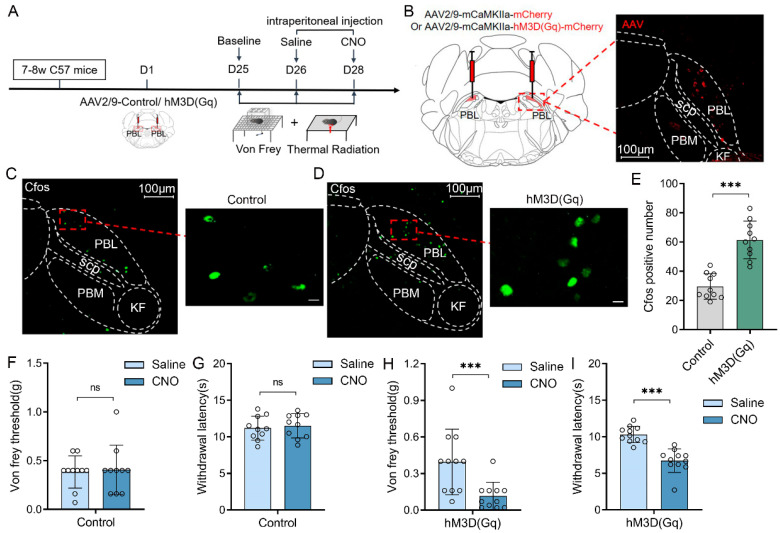
Chemogenetic activation of PBL glutamatergic neurons decreases the pain threshold in mice. (**A**) Flow chart of the behavioural measurements in control and genetically manipulated mice. (**B**) Schematic of virus injection and hM3D(Gq) and mCherry expression in PBL glutamatergic neurons. Scale = 100 μm. (**C**,**D**) C-Fos immunofluorescence (green) in PBL after CNO injection in control (**C**) and hM3D(Gq) (**D**) mice. Scale = 100 μm (**left**) or 10 μm (**right**). (**E**) C-Fos count comparison after CNO injection in control and hM3D(Gq) mice after immunofluorescence staining (n = 10 in both groups). (**F**,**G**) Von Frey threshold and thermal withdrawal latency in control mice after saline or CNO injection (n = 10). (**H**,**I**) Von Frey threshold and thermal withdrawal latency in hM3D (Gq) mice after saline or CNO injection (n = 11). Data are presented as the mean ± SD. *** *p* < 0.001, ns: no significance by a two-tailed independent samples *t* test (**E**–**I**).

**Figure 3 ijms-24-11907-f003:**
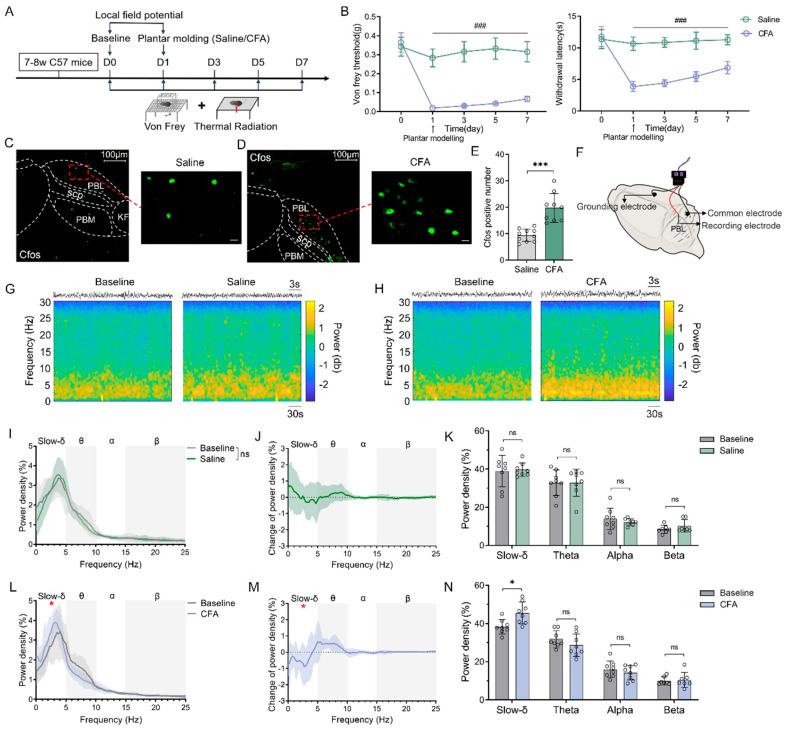
Inflammatory pain excites PBL glutaminergic neurons. (**A**) Flow chart of the behavioural measurements in wild-type mice. (**B**) Von Frey threshold and thermal withdrawal latency in wild-type mice after saline or complete Freund’s adjuvant (CFA) plantar modelling (n = 8–9). (**C**,**D**) C-Fos immunofluorescence (green) in PBL after saline (**C**) and CFA plantar modelling (**D**) in wild-type mice. Scale = 100 μm (**left**) or 10 μm (**right**). (**E**) C-Fos count comparison after saline and CFA plantar modelling in wild-type mice after immunofluorescence staining (n = 9–10). (**F**) Schematic of PBL local field potential electrode implantation. (**G**,**H**) A representative spectrogram of PBL in wild-type mice before and after saline (**G**) or CFA modelling (**H**). (**I**–**K**) Power density of LFP (**I**,**K**) and its group difference (**J**) in slow-delta, theta, alpha, and beta oscillations before and after saline plantar modelling in wild-type mice (n = 8). (**L**–**N**) Power density of LFP (**L**,**N**) and its group difference (**M**) in slow-delta, theta, alpha, and beta oscillations before and after CFA plantar modelling in wild-type mice (* *p* < 0.05 for slow-delta between baseline and CFA group; n = 8). Data are presented as the mean ± SD or mean ± SEM ((**B**), **left**). ^###^
*p* < 0.001 by two-way repeated-measures ANOVA (**B**). * *p* < 0.05, *** *p* < 0.001, ns: no significance by a two-tailed independent samples *t* test (**E**,**I**–**N**).

**Figure 4 ijms-24-11907-f004:**
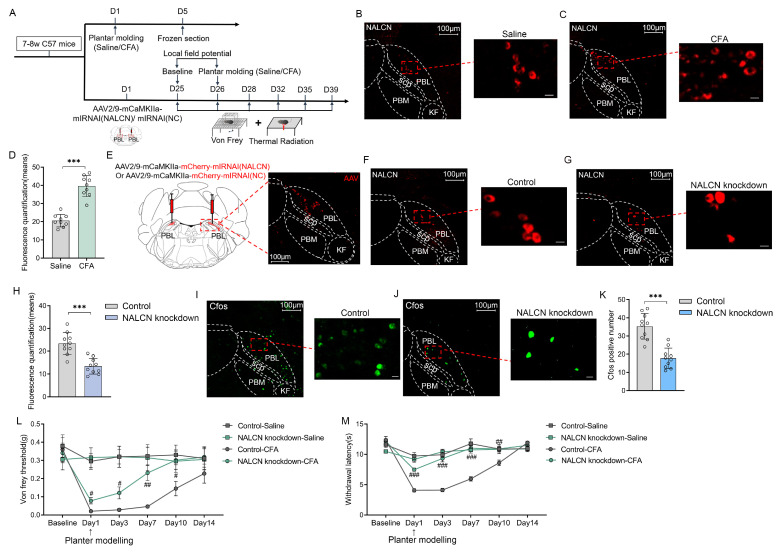
Knockdown of NALCN in PBL glutamatergic neurons alleviates CFA-induced pain. (**A**) Flow chart of virus injection and behavioural measurements in mice. (**B**,**C**) NALCN immunofluorescence (red) after saline (**B**) and CFA plantar modelling (**C**) in wild-type mice after immunofluorescence staining. Scale = 100 μm (**left**) or 25 μm (**right**). (**D**) NALCN fluorescence quantification comparison after saline and CFA plantar modelling in wild-type mice (n = 9 in both groups). (**E**) Injection of control and NALCN knockdown viruses into PBL of C57 6J mice. Scale = 100 μm. (**F**,**G**) NALCN fluorescence (mCherry) in PBL in control (**F**) and NALCN knockdown mice (**G**). Scale = 100 μm (**left**) or 25 μm (**right**). (**H**) NALCN fluorescence quantification comparison in PBL of control and NALCN knockdown mice (n = 10 in both groups). (**I**,**J**) C-Fos fluorescence (green) in PBL in control (**I**) and NALCN knockdown mice (**J**) after CFA plantar modelling. Scale = 100 μm (**left**) or 10 μm (**right**). (**K**) C-Fos count comparison in PBL of control and NALCN knockdown mice before and after CFA plantar modelling (n = 10 in both groups). (**L**) The von Frey threshold in control and NALCN knockdown mice before and after saline or CFA plantar modelling at 14 days (n = 8–11). (**M**) Withdrawal latency in control and NALCN knockdown mice before and after saline or CFA plantar modelling at 14 days (n = 8–11). Data are presented as the mean ± SD or mean ± SEM (**L**). *** *p* < 0.001 by a two-tailed independent samples *t* test (**D**,**H**,**K**). ^###^ *p* < 0.001, ^##^ *p* < 0.01, ^#^
*p* < 0.05 by two-way repeated-measures ANOVA (**L**,**M**).

**Figure 5 ijms-24-11907-f005:**
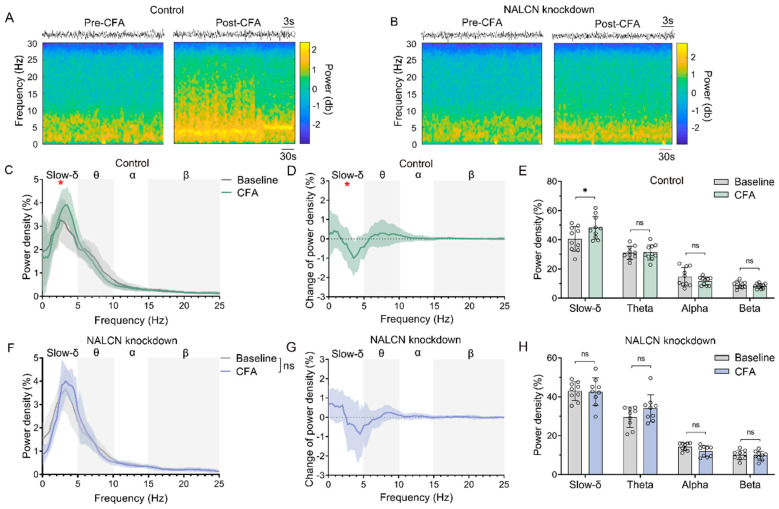
Knockdown of NALCN in PBL glutamatergic neurons inhibits CFA-induced slow-delta oscillations. (**A**,**B**) A representative spectrogram of PBL in control (**A**) and NALCN knockdown mice (**B**) before and after CFA plantar modelling. (**C**−**E**) Power density of LFP (**C**,**E**) and its group difference (**D**) in slow-delta, theta, alpha, and beta oscillations before and after CFA plantar modelling in control mice (* *p* < 0.05 for slow-delta between baseline and CFA group). (**F**−**H**) Power density of LFP (**F**,**H**) and its group difference (**G**) in slow-delta, theta, alpha, and beta oscillations before and after CFA plantar modelling in NALCN knockdown mice. Data are presented as the mean ± SD. * *p* < 0.05, ns: no significance by a two-tailed independent samples *t* test (**C**−**H**).

**Figure 6 ijms-24-11907-f006:**
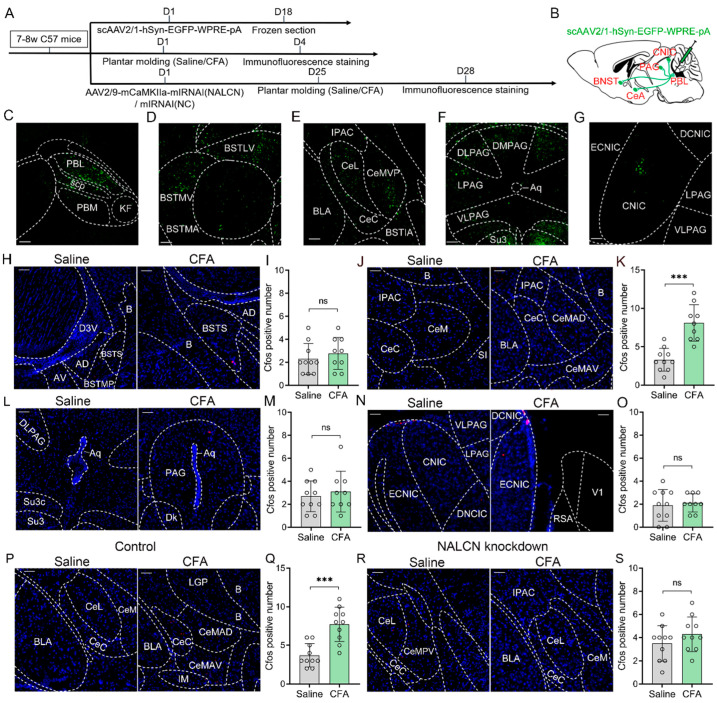
NALCN modulates CFA-induced pain via PBL glutamatergic neuron-central nucleus amygdala (CeA) projections. (**A**) Flow chart of virus injection and related experimental operations. (**B**) Schematic of anterograde tracer virus injection and downstream projection of PBL. (**C**–**G**) After injection of scAAV2/1-hSyn-EGFP-WPRE-pA anterograde tracer virus into the bilateral PBL (**C**), viral fluorescence (EGFP) was distributed in the BNST (**D**), CeA (**E**), PAG (**F**) and CNIC (**G**). Scale = 50 μm. PBL, lateral parabrachial nucleus; scp, superior cerebellar peduncle; PBM, medial parabrachial nucleus; KF, Kölliker-Fuse nucleus; BSTLV, bed nucleus of the stria terminalis, lateral division, ventral part; BSTMV, bed nucleus of the stria terminalis, medial division, ventral part; BSTMA, bed nucleus of the stria terminalis, medial division, anterior part; IPAC, interstitial nucleus of the posterior limb of the anterior commissure; CeL, central amygdaloid nucleus, lateral division; CeMVP, central amygdaloid nucleus, medial posteroventral part; CeC, central amygdaloid nucleus, capsular part; BSTIA, bed nucleus of the stria terminalis, intra-amygdaloid division; BLA, basolateral amygdaloid nucleus, anterior part; DMPAG, dorsomedial periaqueductal gray; DLPAG, dorsolateral periaqueductal gray; LPAG, lateral periaqueductal gray; VLPAG, ventrolateral periaqueductal gray; Su3, supraoculomotor periaqueductal gray; Aq, aqueduct; ECNIC, external central nucleus of the inferior colliculus; CNIC, central nucleus of the inferior colliculus; DCNIC, dorsal central nucleus of the inferior colliculus. (**H**,**J**,**L**,**N**) C-Fos fluorescence (red) and DAPI fluorescence (blue) in the BNST (**H**), CeA (**J**), PAG (**L**) and CNIC (**N**) after saline and CFA plantar modelling in wild-type mice after immunofluorescence staining. Scale = 50 μm. D3V, dorsal 3rd ventricle; B, basal nucleus; BSTS, bed nucleus of stria terminalis, supracapsular part; AD, anterodorsal thalamic nucleus; AV, anteroventral thalamic nucleus; BSTMP, bed nucleus of the stria terminalis, medial division, posterior part; CeM, central amygdaloid nucleus, medial division; SI, substantia innominata; CeMAD, central amygdaloid nucleus, medial division, anterodorsal part; CeMAV, central amygdaloid nucleus, medial division, anteroventral part; Su3, supraoculomotor periaqueductal gray; Dk, nucleus of Darkschewitsch; V1, primary visual cortex; RSA, retrosplenial agranular cortex. (**I**,**K**,**M**,**O**) C-Fos count comparison in the BNST (**I**), CeA (**K**), PAG (**M**) and CNIC (**O**) after saline and CFA plantar modelling in wild-type mice (n = 9–10). (**P**,**R**) C-Fos fluorescence (red) and DAPI fluorescence (blue) in the CeA after saline and CFA plantar modelling in control (**P**) and NALCN knockdown mice (**R**) after immunofluorescence staining. Scale = 50 μm. LGP, lateral globus pallidus; IM, intercalated amygdaloid nucleus, main part. (**Q**,**S**) C-Fos count comparison in the CeA after saline and CFA plantar modelling in control (**Q**) and NALCN knockdown mice (**S**) (n = 10 in both groups). Data are presented as the mean ± SD. *** *p* < 0.001, ns: no significance by a two-tailed independent samples *t* test (**I**,**K**,**M**,**O**,**Q**,**S**).

## Data Availability

Not applicable.

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
