# Peer review of "Sodium Leak Channel in Glutamatergic Neurons of the Lateral Parabrachial Nucleus Modulates Inflammatory Pain in Mice"

_ijms, 2023, doi:10.3390/ijms241511907_

Round 1

Reviewer 1 Report

Comments and Suggestions for Authors

The manuscript (ijms-2512606) by Wu et al contains interesting perspectives of glutamatergic neurons in the lateral parabrachial nucleus (PBL) are associated with the pathogenesis of inflammatory pain. However, I have a few concerns which are listed below for the authors to improve this manuscript –

Major Concerns-

1-    In the figure 1, author evaluated the pain threshold by chemogenetic inhibition of PBL glutaminergic neurons after injecting clozapine N-oxide (CNO) dissolved in the 0.9% saline. So, saline is the proper control to compare with.

a.     What was the rationale to use baseline data over saline to compare with CNO data?

b.     Data presentation with baseline, saline and CNO groups seems confusing as authors applied student’s t-test. In addition to this, baseline data is not a proper control for statistical analysis. Authors are suggested to reanalyze the data and present it as saline vs CNO.

c.     Authors are suggested to use baseline data comparisons as supplementary, in case if it is important.

2-    Similarly rest of all figures (2-6) needed to be updated by using proper control.

Reviewer 2 Report

Comments and Suggestions for Authors

My suggestions, concerns,

1. Are the mutations in the NALCN gene associated with any kind of neurological disease, especially inflammation-related diseases? Authors may mention it briefly. 

2. I would add the flowcharts of experiments in the Methods section (Figure 1A, Figure 2A. Figure 3a, Figure 4A). The authors may mention how many wild-type and C57BL/6J mice were used for the experiments. 

3. In the discussion, I would add a pathway, on how NALCN contributes to inflammatory pain

4. Were the differently expressed genes examined in the case of NALCN knockdown in the mice? 

Reviewer 3 Report

Comments and Suggestions for Authors

Review on the manuscript of Wu L et al.: “Sodium Leak Channel in Glutamatergic Neurons of the Lateral Parabrachial Nuclei Modulates Inflammatory Pain in Mice”.

In this manuscript, authors explored the involvement of glutamatergic neurons located in the parabrachial lateral nucleus (PBL) in the development of inflammatory pain in mice. Authors show, by chemogenetic activation and inhibition, that PBL glutamatergic neurons are involved in the transmission of pain signals, and that this sensory transduction involves sodium leak channels (NALCN). In addition, authors show that NALCN may modulate inflammatory pain via PBL glutamatergic neuron-central nucleus amygdala projections.

The manuscript is very clear and well written. In addition, I consider that authors made a great job with the figures, as they are very organized and very clear to the readers. The issues that arise to me are listed below, so, I hope the authors find the following comments and suggestions useful.

1 – I consider that authors made a great job with the figures, as they are very organized and, in general, clear to the readers. However, I recommend authors to increase the font size of panel A for figures 3, 4 and 6.

2 – Regarding figure 6, in panels C-G, H, J, L and N there are several nuclei identified. To make the information clear to the readers, I recommend authors to show at the end of the figure legend what each abbreviation of the nuclei means.

3 – I recommend authors to provide the ID of the license of the study approval by the Animal Ethics Committee of the West China Hospital of Sichuan University.

4 – What are the authors’ expectations about this mechanism of inflammatory pain regulation in female mice? Would it be the same as authors show for male mice? And the age, would aging affect the regulation of inflammatory pain by this mechanism?

Round 2

Reviewer 1 Report

Comments and Suggestions for Authors

The manuscript has been improved after revision. Thank you for implementing the suggestions. No further comments.

Reviewer 2 Report

Comments and Suggestions for Authors

Thank you, the authors fulfilled my suggestions.